# Clinical and imagenologic significance of the neutrophil-to-lymphocyte ratio in neuromyelitis optica spectrum disorder: A systematic review with meta-analysis

**Miguel Cabanillas-Lazo**[1,2], **Claudia Cruzalegui-Bazán**[1,3], **Milagros Pascual-Guevara**[1,3], **Carlos Quispe-Vicuña**[1,2], **Fernando Andres Terry-Escalante**[2,4], **Nicanor Mori**[5], **Carlos Alva-Díaz**[2,5,6]*

1 Sociedad Cientifica de San Fernando, Lima, Peru, 2 Red de Eficacia Clinica y Sanitaria (REDECS), Lima, Peru, 3 Facultad de Medicina, Universidad Nacional Mayor de San Marcos, Lima, Peru, 4 Facultad de Medicina Humana, Universidad de San Martin de Porres, Lima, Peru, 5 Servicio de Neurología, Departamento de Medicina y Oficina de Apoyo a la Docencia e Investigación (OADI), Hospital Daniel Alcides Carrión, Callao, Perú, 6 Universidad Señor de Sipán, Chiclayo, Perú

* alvacarl@crece.uss.edu.pe

## Abstract

### Background

Recently, the neutrophil-lymphocyte ratio (NLR) has become a biomarker for assessing inflammatory stress and prognosis in different diseases.

### Objective

We aimed to conduct a systematic review and meta-analysis to summarize the current evidence on the capacity of the NLR to serve as a biomarker in neuromyelitis optica spectrum disorder (NMOSD).

### Methods

Through a comprehensive systematic search up to December 2021 and using the search terms "neutrophil-to-lymphocyte ratio" and "neuromyelitis optica spectrum disorder" we selected studies evaluating NLR values in NMOSD patients. A meta-analysis was planned, and a narrative synthesis was performed when this was not possible. Subgroup and sensitivity analyses were planned. The Grading of Recommendations, Assessment, Development and Evaluations (GRADE) approach was used to assess certainty of the evidence.

### Results

Six studies were included (1036 patients). A significant increase in the NLR was observed between NMOSD patients and healthy controls with high heterogeneity (MD: 1.04; 95% CI: 0.76; 1.32; $I^2$ = 59%). Regarding NMOSD prognosis, relapse (OR: 1.33 –OR: 2.14) was evaluated as being related to NLR with low certainty. An association with Expanded Disability Status Scale (EDSS) score $\geq$4 (OR: 1.23 –OR: 1.43) was reported with moderate

**Data Availability Statement:** All relevant data are within the paper and its Supporting Information files.

**Funding:** The author(s) received no specific funding for this work.

**Competing interests:** The authors have declared that no competing interests exist.

certainty. An association with the occurrence of lesions on MRI was reported with an OR of 1.52.

## Conclusion

We found the NLR to be useful as a biomarker of NMOSD as it was significantly increased in the patient group compared to the healthy control group with high certainty. Additionally, the NLR was applicable as an indicator of poor prognosis with low to moderate certainty.

## Introduction

Neuromyelitis optica spectrum disorder (NMOSD) is an inflammatory autoimmune disease of the central nervous system (CNS) characterized by optic neuritis, myelitis and cerebral or brainstem syndromes [1]. Although it is an uncommon pathology with a prevalence of 0.5 per 100,000 inhabitants, NMOSD can increase sixfold in certain racial groups, such as blacks and Asians [1], who are younger at the onset of the disease and develop more NMOSD complications, such as permanent visual impairment and, to a lesser extent, permanent motor disability and wheelchair dependence [2].

Some studies postulate the participation of circulating neuromyelitis optica immunoglobulin G autoantibodies directed against the astrocytic endfeet of AQP-4 [3]; these autoantibodies generate neuronal damage by activating the membrane attack complex, leading to an inflammatory reaction characterized by the presence of neutrophils, eosinophils and macrophages in high proportion but lymphocytes in small proportion [4, 5]. According to a study in which antineutrophil IgG was injected into a group of mice, it was found that in nonneutropenic mice compared to neutropenic mice with greater gravity, there was a larger number of inflamed vessels with adhered neutrophils both luminally and perivascularly that expressed significant loss of myelin and AQP-4. In addition, half of these were active or degranulated neutrophils present in the migration phase, so their presence would be associated with an acute phase of NMOSD [6]. According to another study in which seven patients were analyzed, neutrophils contributed to the pathogenesis of this disease due to their participation in the deregulation of astrocytic function. This occurs through the formation of reactive astrocytes, which contribute to inflammatory processes [7]. Additionally, neutrophils in NMOSD show reduced adhesion and migratory capacity as well as decreased production of reactive oxygen species and degranulation [8].

Therefore, this elevated neutrophil and low lymphocyte participation can be expressed in the neutrophil-lymphocyte ratio (NLR), which, according to recent studies, could be a novel and clinically relevant biomarker used to indicate a higher risk of depression and to predict poor 3-month functional outcomes in ischemic stroke patients [9, 10].

Against this background, the aim of this systematic review is to summarize the current knowledge regarding the capacity of the NLR to serve as a biomarker in NMOSD.

## Materials and methods

This systematic review was reported according to the Preferred Reporting Items for Systematic Reviews and Meta-Analyses (PRISMA) [11]. See S1 Checklist, PRISMA checklist. The study protocol was registered in PROSPERO with the CRD42022306366.

## Data sources

We searched PubMed, Embase, Scopus, Web of Science, SciELO Citation Index, and Google Scholar up to December 2021 using the search terms "neutrophil-to-lymphocyte ratio" and "neuromyelitis optica spectrum disorder" The search strategy for PubMed was adapted for use in the other databases (**S1 Table**). There were no restrictions on language or publication date. We completed the search by reviewing the bibliographic references of the included studies and selecting the articles that met the requirements.

## Eligibility criteria

Studies were included if they met the following criteria: 1) studies included adult participants (aged > 18 years old); 2) NLR values were assessed during the pretreatment period in NMOSD patients or controls without NMOSD; and 3) analytical observational studies (cross-sectional, case-control and cohort studies). We excluded narrative and systematic reviews, studies in nonhumans, case reports, conference abstracts and letters.

## Study selection

The electronic search results were imported into Endnote X9, and duplicate records were exported to Rayyan (https://rayyan.qcri.org/). The selection based on the title and abstract was performed by two reviewers (CQV and FTE), and any discrepancies were resolved by consensus and consideration of the opinion of the third reviewer (CAD). These reviewers assessed the inclusion criteria independently by reading the full texts of the potentially relevant studies that were selected, and discrepancies were resolved according to consensus. The complete list of excluded articles is provided in **S2 Table**.

## Outcomes

Our outcomes were as follows: 1) evaluation of NLR (calculated as a simple ratio between the neutrophil and lymphocyte counts measured in peripheral blood) [12] comparing NMOSD patients (diagnosed according to International Diagnostic Criteria Neuromyelitis Optica Spectrum Disorders) [13] with healthy control subjects; and 2) evaluation of NLR at the time of occurrence of at least one of the following events after discharge: 1) relapse (defined as appearance of neurological symptoms or exacerbation of existing neurological symptoms lasting for >24 h, time to the last onset of >1 month, and imaging confirmation of a new lesion); 2) appearance of new/enlarging and/or enhancing lesion on Magnetic Resonance Imaging (MRI); and 3) severity assessment using the Extended Disability Status Scale (EDSS) [14] (which rates the disability status of MS patients on a scale of 0 to 9) with emphasis on level 4.0 (Relatively Severe Disability) and level 7.0 (Essential Wheelchair Restriction).

Typical NMOSD lesions on brain MRI were localized on the optic nerve, brainstem/cerebellum, area postrema, diencephalon and spinal cord (longitudinally extensive transverse myelitis, short-segment transverse myelitis and multisegmented).

## Data extraction

Two authors (CCB and FTE) independently carried out data extraction using a data extraction form, and any disagreements were resolved by consensus and, ultimately, by a third author (CAD). We extracted the following information: title of the study, first author, year of publication, study design, country where the study was performed, number of participants, sex, age, sample time, mean or median NLR with standard deviation (SD) according to sample stratification, crude and adjusted association measures, type of outcome and its definition. If

additional data were needed, we contacted the corresponding author through email to request further information.

### Risk of bias assessment

The quality of the studies was assessed with the Newcastle Ottawa Scale (NOS) [15] by two authors (CCB and FTE). This tool evaluates the quality of published nonrandomized studies and is based on three items: selection, comparability, and outcome/exposure. Each item has subitems to which a star-based score is assigned. Studies with scores $\geq 6$ were considered to have a low risk of bias (high quality); those with scores of 4–5 were considered to have a moderate risk of bias; and those with scores of $< 4$ were considered to have a high risk of bias.

### Statistical analysis

A meta-analysis was planned for each outcome; however, when this was not possible due to unavailable data, narrative synthesis was performed. Meta-analysis was performed using a random-effects model. The variance between studies ($\tau2$) was estimated using the DerSimonian-Laird estimator. Mean differences (MD) with their 95% confidence intervals (CI 95%) between NMOSD patients and controls were pooled. Heterogeneity between studies was assessed using the $I^2$ statistic. Heterogeneity was defined as low if $I^2$ was $<30\%$, moderate if $I^2$ was = 30–60%, and high if $I^2$ was $> 60\%$. The metacont function of the metapackage in R 4.1.0 was used (www.r-project.org). Finally, we performed a subgroup analysis to identify potential sources of heterogeneity and leave-one-out sensitivity analysis.

### Evidence certainty assessment

Two authors (CCB and CQV) independently assessed the certainty of our pooled results and qualitative synthesis by applying the Grading of Recommendation, Assessment, Development, and Evaluation (GRADE) system of rating to continuous outcomes [16] and narrative synthesis [17]. This assessment is based on five domains: study limitations (risk of bias of the studies included), imprecision (sample size and confidence interval), indirectness (generalizability), inconsistency (heterogeneity), and publication bias as stated in the GRADE handbook and GRADE for prognostic factors [18] and in the GRADE adaptation for assessment of evidence about prognosis [19]. We adapted the assessment to our results. The certainty of the evidence was characterized as high, moderate, low, or very low.

## Results

### Study selection

We identified 376 studies through our systematic search. We removed duplicates and screened 308 studies. Finally, we included 6 articles [20–25] (**Fig 1**).

### Characteristics of studies

All the included studies involved retrospective cohorts. The total number of participants was 1527. A total of 806 participants were NMOSD patients, and 721 were healthy controls. The average age range of NMOSD patients was between 36.29 and 48.71. The summary of the characteristics of the studies is summarized in **Table 1**.

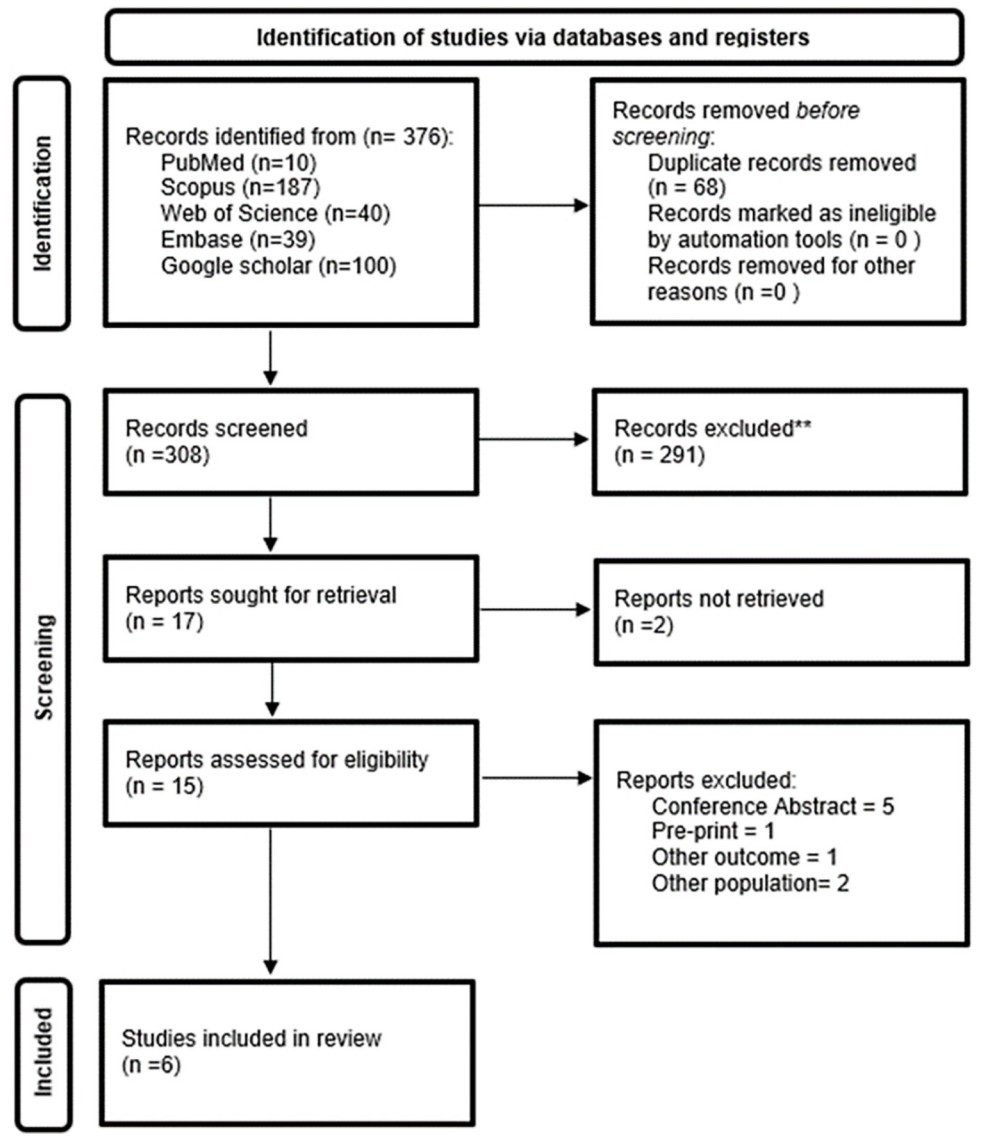

**Fig 1. PRISMA flowchart of included studies.**

## Risk-of-bias assessment

We assessed the risk of bias of each of the 6 included studies. Five articles had a "low risk of bias." The study by Lin et al. [22] obtained the lowest score (5/9); the paper did not mention the calculation of the sample, there was no control of confounding variables and adequate follow-up of the cohorts was missing (S3 Table).

## NMOSD patients and healthy controls

Among the 6 studies that met our inclusion criteria, three were pooled. A total of 1036 participants were selected, of which 382 were assigned to the NMOSD group and 654 to the control group. In the pooled analysis, a significant increase in the NLR was observed between groups with moderate heterogeneity (MD: 1.04; 95% CI: 0.76; 1.32; $I^2$ = 59%) (Fig 2).

**Table 1. Characteristics of the included studies evaluating the clinical and imagenologic significance of the neutrophil-to-lymphocyte ratio (NLR) in neuromyelitis optica spectrum disorder (NMOSD) (n = 6).**

| Study-ID | Country | Study design | Main inclusion criteria | Main exclusion criteria | N NMOSD (% AQP4-IgG +) | N Controls | Match NMOSD—control | NLR values at onset: NMOSD Control | | Population characteristics: Age (years) mean | Blood-sampling time | Follow-up time |
|---|---|---|---|---|---|---|---|---|---|---|---|---|
| Carnero, 2021 [20] | Argentina, Ecuador, México | Retrospective cohort | First NMOSD attack from 18 to 80 years, without treatment | Autoimmune, liver, cardiometabolic, malignancies and hematologic conditions | 90 (100) | 365 | NR | 2.9 ± 1.6[1] | | **Male sex (%):** 16.7% | Day post admission | 24 months |
| | | | | | | | | 1.8 ± 0.7[1] | | **Age:** 42.3 (14.3)[1] | | |
| Chen, 2021 [21] | China | Retrospective cohort | NMOSD AQP4-IgG + | MOG-IgG+ | 32 (100) | 31 | NR | 2.54[2] | | **Male sex (%):** 3.8 | Before immunotherapies | NR |
| | | | | | | | | | | **Age:** 45.5 (19–69)[2] | | |
| Lin, 2017 [22] | China | Retrospective cohort | NMOSD patients | Autoimmune disorders; malignant, cardiovascular, renal, liver and hematology diseases | 46 (92.2) | 120 | NR | 2.55 ± 1.56[1] | | **Male sex (%):** 3.36 | Before treatment | NR |
| | | | | | | | | 1.56 ± 0.53[1] | | **Age:** 36.29 (11.21)[1] | | |
| Xie, 2021 [23] | China | Retrospective cohort | First NMOSD attack | Autoimmune, liver, kidney, cardiovascular diseases, malignancy; other neurological disease; loss of blood cells; children under 6 years old; and previous steroid treatment. | 324 (55.5) | NR | NR | 2.59 (1.63–4.28) | | **Male sex (%):** 28.7 | Day after admission | 43 months |
| | | | | | | | | | | **Age:** 43.40 (17.03)[2] | | |
| Zhou, 2021 [24] | China | Retrospective cohort | First NMOSD attack | Other diseases affecting the Expanded Disability Status Scale (EDSS) score; hematological disease, the active period of other chronic infectious diseases or other conditions that could affect the blood count | 259 (61.6) | 169 | Age | 2.54 (1.72–4.27)[2] | Sex | **Male sex (%):** 17.4 | 12 hours post admission | NR |
| | | | | | | | Smoking history | | | | | |
| | | | | | | | 1.73 (1.36–2.24)[2] | | History of alcohol consumption | **Age:** 45 (30–56)[2] | | |
| Yangyan, 2021 [25] | China | Retrospective cohort | NMOSD patients | Cardiovascular and cerebrovascular diseases, malignant tumors, coagulation disorders and kidney failure | 55 (23) | 36 | NR | Age | | **Male sex (%):** 6.59 | Day after admission | NR |
| | | | | | | | | | | **Age** | | |
| | | | | | | | | Sex | | **NMOSD group:** 48.71 (14.41)[1] | | |
| | | | | | | | | | | **Control group:** 53.81 (10.43)[1] | | |

[1] Mean±standard deviation

[2] Median (IQR)

NR: Not Reported

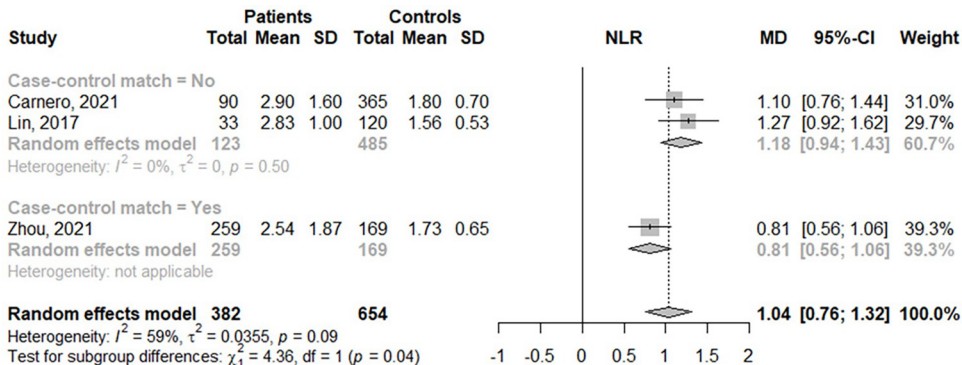

**Fig 2. Mean difference in the neutrophil-to-lymphocyte ratio (NLR) between NMOSD patients and healthy controls according to reported match.**

Regarding subgroup analysis between studies with matched patients and controls, we observed that NMOSD patients had a higher NLR than healthy controls in matched studies (MD: 1.18; 95% CI: 0.94; 1.43; $I^2$ = 86%) (MD: 0.81; 95% CI: 0.56; 1.06; $I^2$ = 0%) (Fig 2).

Regarding sensitivity analysis, when single studies were sequentially removed, no variation in the pooled MD was observed, with an effect size of 1.04 in both. This suggests that the results of the meta-analysis were stable (Fig 3).

## NMOSD prognosis

Xie et al. and Carnero et al. reported an OR of 1.33 to 2.14 for the relapse outcome. For the EDSS≥4 outcome, an OR of 0.96 to 1.23 was observed. However, these were not determined by meta-analyses because different NLR cutoff points were involved. At the same time, Carnero et al. also reported lesions on MRI as an outcome with an OR of 1.52 (95% CI: 1.14; 2.03) (Table 2).

## Evidence certainty

We used GRADE for continuous outcomes in our meta-analysis of NMOSD patients and healthy controls (Table 3). GRADE for narrative synthesis was used in assessing the clinical and radiographic outcomes of NMOSD. We did not downgrade for inconsistency even though $I^2$ was 59% because our subgroup analysis showed that the problem could be due to the case-control match.

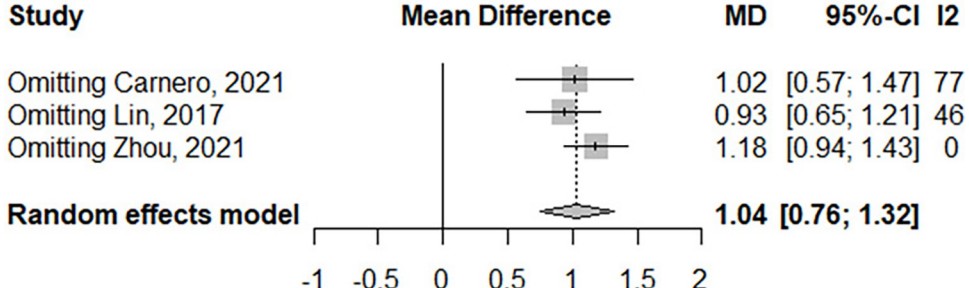

**Fig 3. Mean difference in the neutrophil-to-lymphocyte ratio (NLR) between NMOSD patients and healthy controls according to the exclusions of each study.**

**Table 2. Characteristics of the included studies evaluating the prognostic use of the neutrophil-to-lymphocyte ratio (NLR) for outcomes in neuromyelitis optica spectrum disorder (NMOSD).**

| Study | Number of analyzed patients | Outcome | Association between NLR and poor outcome | NLR cutoff | Follow-up |
|---|---|---|---|---|---|
| **Xie, 2021** [23] | 324 | Relapse | OR: 1.33 (1.11–1.59) | 2.38 | 43 months |
| | | EDSS score (≥4) | OR: 1.23 (1.06–1.43) | 2.63 | |
| **Carnero, 2021** [20] | 90 | Relapse | OR: 2.14 (1.35–3.39) | NR | 24 months |
| | | EDSS score (≥4) | OR: 0.96 (0.64–1.43) | NR | |
| | | Lesions on MRI* | OR: 1.52 (1.14–2.03) | NR | |

NR: Not reported; OR: Odds ratio. EDSS: Expanded Disability Status Scale; NLR: Neutrophil-to-lymphocyte ratio; MRI: Magnetic resonance imaging. *: Defined as new/enlarging and/or contrast enhancing lesion (s) on MRI

Concerning NMOSD relapse, we judged the certainty as low. We started with a rating of high certainty because all the studies included were cohorts, and we downgraded them according to imprecision and inconsistency. For an EDSS score ≥4, we judged the certainty as moderate. We started with a rating of high certainty because all the studies included were cohorts, and we downgraded because one study was not statistically significant.

Finally, for new lesions, MRI was judged as having moderate certainty. We started with a rating of high certainty because the unique study was a cohort, and we downgraded it according to imprecision (OR: 1.52 [1.14–2.03]) (**Table 4**).

## Discussion

### Summary of main results

This systematic review with meta-analysis (1527 participants included) revealed a higher NLR average in NMOSD patients than in healthy controls with high certainty. Furthermore, we found that patients with poor long-term prognoses (NMOSD relapse, moderate incapacity, and new lesions on MRI) had higher NLR values than those who had better results.

### NLR in NMOSD patients versus healthy controls

Our pooled analysis showed that the mean NLR was significantly higher in NMOSD patients than in healthy controls. These results were not weakened by any of the studies that reported

**Table 3. GRADE summary of findings between NMOSD patients and healthy controls.**

| Outcomes | of participants | Certainty of the evidence | Anticipated absolute effects | |
|---|---|---|---|---|
| | (studies) | (GRADE) | Mean Difference (MD) | 95% CI |
| Difference between NMOSD patients and healthy participants | 1527 | | 1.04 | 0.76–1.32 |
| | (6 observational studies) | High[a] | | |

**CI:** confidence interval; **MD:** mean difference

**GRADE Working Group grades of evidence**

**High certainty:** We are very confident that the true effect lies close to that of the estimate of the effect.

**Moderate certainty:** We are moderately confident in the effect estimate; the true effect is likely to be close to the estimate of the effect, but there is a possibility that it is substantially different.

**Low certainty:** Our confidence in the effect estimate is limited; the true effect may be substantially different from the estimate of the effect.

**Very low certainty:** We have very little confidence in the effect estimate; the true effect is likely to be substantially different from the estimate of the effect.

Explanations

a. There was high certainty even though I2 = 59% because our subgroup analysis showed that the reason could be case-control matching.

**Table 4. GRADE summary of findings of prognostic outcomes in NMOSD patients.**

| Outcomes | No. of participants | Certainty of the evidence | Anticipated absolute effects | |
|---|---|---|---|---|
| | (studies) | (GRADE) | Mean Difference (MD) | 95% CI |
| Relapse | 766 | | 1.33 | 1.11–1.59 |
| | (2 observational studies) | Low[a,b] | | |
| EDSS≥4 | 766 | | 0.96 | 0.64–1.43 |
| | (2 observational studies) | Moderate[c] | | |
| New lesions of MRI | 455 | | 1.52 | 1.14–2.03 |
| | (1 observational study) | Moderate[d] | | |

**CI:** confidence interval; **MD:** Mean difference; **OR:** Odds ratio; **EDSS:** Expanded Disability Status Score; **MRI:** Magnetic resonance imaging

**GRADE Working Group grades of evidence**

**High certainty:** We are very confident that the true effect lies close to that of the estimate of the effect.

**Moderate certainty:** We are moderately confident in the effect estimate; the true effect is likely to be close to the estimate of the effect, but there is a possibility that it is substantially different.

**Low certainty:** Our confidence in the effect estimate is limited; the true effect may be substantially different from the estimate of the effect.

**Very low certainty:** We have very little confidence in the effect estimate; the true effect is likely to be substantially different from the estimate of the effect.

Explanations

a. Odds ratio difference greater than 0.5 between studies

b. The confidence interval of the larger study crossed 1.25

c. The confidence interval of one study crossed no effect point.

d. The confidence interval passed 1.25 in the unique study.

case-control matching as the sensitivity analysis showed. This is consistent with previous reviews that also evaluated the NLR in other autoimmune diseases. Wang et al. [26] reported in a meta-analysis of 14 studies that higher NLR values (standardized mean difference [SMD] = 1.43; 95% CI 0.98–1.88; p<0.001) were present in patients with systemic lupus erythematosus than in healthy patients. In addition, Olsson et al. [27] analyzed four case-control studies and found higher NLR values in MS patients than in healthy controls. Finally, Paliogiannis et al. [28] and Erre et al. [29] reported similar results for psoriasis (SMD = 0.69; 95% CI 0.53–1.85; p<0.001) and rheumatoid arthritis (SMD = 0.79; 95% CI 0.55–1.03; p< 0.001). This would show that high NLR values could be used to differentiate between patients with autoimmune diseases and healthy persons. The mechanism involved in this differentiation could be an imbalance between the number of neutrophils representing innate immune cells and the number of lymphocytes representing acquired immune cells. This is due to the alteration of the permeability of the blood-brain barrier in neurological diseases [30].

## NLR as a prognostic factor in NMOSD patients

This review found the NLR to be a prognostic biochemical marker for relapse, EDSS score ≥4 and presence of lesions by magnetic resonance imaging in patients with NMOSD; this finding is in line with the results of other systematic reviews of patients with MS since the NLR was higher in patients with relapses than in healthy individuals or those with disease in remission [27]. In addition, two of the studies reported cutoff points of 4.52 (sensitivity: 96.1%; specificity: 42.9%) and 3.9 for predicting worse progression of disability assessed as EDSS ≥5 and EDSS >3, respectively. There were no other SRs for NLR and autoimmune neurological diseases; however, there were cohorts in patients with dermatomyositis/polymyositis [31] and systemic lupus erythematosus [27] where NLR was predictive of the overall mortality mainly due to pulmonary complications (cutoff 4.78 and hazard ratio [HR]: 5.20; CI: 1.92–14.07) and development of lupus nephritis (SMD: 0.77; CI: 0.57–0.97). Finally, in pregnant women

diagnosed with AQP4-positive NMOSD, an NLR of 5.25 (sensitivity: 72.7%; specificity: 90.0%) was found to predict a pregnancy-related NMOSD attack defined as relapse, onset or worsening of neurological signs, or the presence of a newly enhanced lesion on MRI [32].

### Recommendations for future research

The certainty of the evidence for the NLR between NMOSD patients and healthy controls was high. Our subgroup analysis showed that the heterogeneity could be explained by case-control matching. This could mean that age, gender, or other variables should be identified as confounding variables, so they should be considered in future studies. Otherwise, the certainty of the prognosis was low to moderate due to inconsistency and imprecision. Therefore, prospective studies with appropriate follow-up times and sample sizes are required to determine the precision of the results and validate the NLR cutoff points. In addition, the studies that evaluated prognosis represented South America and China, so we recommend conducting studies in other populations due to heterogeneity in prevalence and clinical manifestations between regions and ethnic groups [33]. Finally, we recommend the development of prognostic modeling studies that consider clinical and biochemical variables such as NLR whose values are easy and inexpensive to obtain and which may be used to predict adverse outcomes with high accuracy in these patients.

### Clinical applicability

The NLR is a biomarker of systemic inflammation that has also been shown to be an early diagnostic indicator in cancer [34] and neuroimmune diseases such as MS [27]. Our results indicate that the NLR could serve to both differentiate NMOSD patients from healthy patients and indicate poor long-term prognosis. The latter would support the idea of using the NLR once the diagnosis is established. Otherwise, other possible NMOSD biomarkers have been reported, such as serum levels of cytokines/chemokines and neurofilament light chain [35]. However, all these markers require equipment and advanced preparation for utilization. In this context, the NLR gains clinical importance since it is derived from tests that are simple to perform, inexpensive and routinely available and can be used in conjunction with other clinical or laboratory variables to generate valid predictive models.

### Limitations and strengths

Our systematic review has some limitations. First, most of the studies were retrospective in nature, so their results could be susceptible to confounding factors. Second, most studies were conducted in China, which could affect the generalizability of our results. In addition, there was heterogeneity and imprecision in our prognosis outcomes. Despite everything mentioned, this article also has strengths. First, it is innovative in the topic it addresses since it is the first systematic review with meta-analysis that evaluates the prognostic value of the NLR in NMOSD. There were also no restrictions at the time of executing the bibliographic search, so the possibility of publication bias is low. Likewise, we evaluated the certainty of our results with the GRADE approach.

### Conclusions

Based on our results, with high certainty, the mean NLR is higher in NMOSD patients than in healthy controls. Furthermore, with low to moderate certainty, we found that the NLR could be a prognostic factor for relapse, disability (EDSS$\geq$4), and appearance of new lesions on MRI.

Future prospective studies in different populations and development of prognostic models that take the NLR into account are needed.

## Supporting information

**S1 Checklist. PRISMA checklist.**
(DOCX)

**S1 Table. Search strategy.**
(DOCX)

**S2 Table. Excluded studies.**
(DOCX)

**S3 Table. Newcastle—Ottawa quality assessment scale for included studies.**
(DOCX)

## Author Contributions

**Conceptualization:** Miguel Cabanillas-Lazo, Nicanor Mori, Carlos Alva-Díaz.

**Data curation:** Claudia Cruzalegui-Bazán, Milagros Pascual-Guevara, Carlos Quispe-Vicuña, Fernando Andres Terry-Escalante.

**Formal analysis:** Miguel Cabanillas-Lazo.

**Funding acquisition:** Carlos Alva-Díaz.

**Investigation:** Miguel Cabanillas-Lazo, Claudia Cruzalegui-Bazán, Milagros Pascual-Guevara, Carlos Quispe-Vicuña, Fernando Andres Terry-Escalante.

**Methodology:** Miguel Cabanillas-Lazo, Nicanor Mori, Carlos Alva-Díaz.

**Project administration:** Miguel Cabanillas-Lazo, Claudia Cruzalegui-Bazán.

**Resources:** Miguel Cabanillas-Lazo, Nicanor Mori, Carlos Alva-Díaz.

**Software:** Miguel Cabanillas-Lazo.

**Supervision:** Nicanor Mori, Carlos Alva-Díaz.

**Validation:** Nicanor Mori, Carlos Alva-Díaz.

**Visualization:** Nicanor Mori, Carlos Alva-Díaz.

**Writing – original draft:** Miguel Cabanillas-Lazo, Claudia Cruzalegui-Bazán, Milagros Pascual-Guevara, Carlos Quispe-Vicuña, Fernando Andres Terry-Escalante.

**Writing – review & editing:** Nicanor Mori, Carlos Alva-Díaz.

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
