## [Decision Letter · Decision Letter 0]

6 Nov 2022

PONE-D-22-23904Clinical and imagenologic significance of neutrophil-to-lymphocyte ratio in neuromyelitis optica spectrum disorder: a systematic review with meta-analysisPLOS ONE

Dear Dr. Alva Diaz,

Thank you for submitting your manuscript to PLOS ONE. After careful consideration, we feel that it has merit but does not fully meet PLOS ONE’s publication criteria as it currently stands. Therefore, we invite you to submit a revised version of the manuscript that addresses the points raised during the review process.

We look forward to receiving your revised manuscript.

Kind regards,

Ralf A. Linker, MD

Academic Editor

PLOS ONE

Journal Requirements:

Reviewers' comments:

Reviewer's Responses to Questions

**Comments to the Author**

1. Is the manuscript technically sound, and do the data support the conclusions?

Reviewer #1: Partly

Reviewer #2: Yes

2. Has the statistical analysis been performed appropriately and rigorously? 

Reviewer #1: No

Reviewer #2: Yes

3. Have the authors made all data underlying the findings in their manuscript fully available?

Reviewer #1: No

Reviewer #2: Yes

4. Is the manuscript presented in an intelligible fashion and written in standard English?

Reviewer #1: No

Reviewer #2: Yes

5. Review Comments to the Author

Reviewer #1: The authors address an interesting question on a simple biomarker in NMOSD. While the design of the study seems appropriate in the first place, there a several issues about the methods and the results. A part of these issues might be related to several problems with the English language that needs a thorough revision. More importantly, I am not sure if the modeling and the interpretation of the results is correct. It looks like the main finding is not significant despite the author state that it is significant. I suggest review the models and the reporting (e.g. 95%CI) with a statistician. Please find below some detailed comment for the introduction, methods and the beginning of the result section. However, due to the points mentioned above, I stopped to comment further.

Abstract :

The method section seems a little narrative – please be more specific. For example add search terms and time. GRADE should either be explained or replaced by an appropriate description. The results indicate that the quantitative analysis was possible and the phrase on the narrative part is not needed.

Results: Its not clear which what kind of group the comparison was performed. Please review the wording it is not always very clear. For example: “A significant increase in NLR was observed between NMOSD patients with high heterogeneity” –Between what NMOSD and who ?

Introduction:

Please revise the phrase “According to Kitley et al [2], treated patients developed complications and relapses months later compared to the group treated late.” The point is to underline the efficacy and need for an early treatment?

I disagree twithe the phrase “The pathophysiological mechanism of NMOSD is not completely understood, but some studies postulate the participation of circulating neuromyelitis optica immunoglobulin G autoantibodies directed against the astrocytic endfeet of AQP-4”. I think AQP-4-AB are widely accepted in their central pathophysiological role. The uncertainty is more related to AQP-4 negative patients…

The chain of arguments from AQP-4 to neutrophiles in NMOSD is not very clear and needs a revision. Start with the basic mechanism, summarize current knowledge on neutrophiles etc. Add the mouse results later on.

Expanding a little more on the NLR ratio would be helpful – especially if there is evidence from other diseases that underline the usefulness of this parameter (e.g. NLR is predictive for disease activity in condition X and is treatment response marker in disease Z…).

Methods:

Please add search terms to the method section.

Did you screen abstracts before going into full texts?

Was English language a criterion?

The restriction to treatment naïve patients should be mentioned in the abstract

The wording “peer review” seems a little strange in this context.

In the outcome section you state that you extracted NLR values compared with healty controls but this is contradictory to the inclusion criterion cohort study.

I do not understand the definition of NMOSD severe and mild. Is this based on disability or disease activity? How do relapses enter here? Moreover, I am not sure what the meaning of the MRI lesion criterion is. This looks rather like a MS criterion. The definition of the EDSS can be shortend and I would suggest defining only important milestones such as EDSS 4 and 7 for example.

What is “follow-up crude and adjusted association measures” ?

Did you extract SD ? I think you need the variance for metaregression.

While statistical approach seems reasonable please review the wording. You probably used “random effects models”? Moreover, please specify your model. You used the mean difference in NLR as outcome? Maybe it would be better to use the NLR values from patients and controls. This relates also to the question if healthy control cohorts are well matched or a source of variance.

Were models adjusted for covariates as rate of AQP-4 positivity, age and Sex?

Results

The NLR values are not reported in Table 1 but should be added. Also the corresponding values from controls.

The two stars in Figure 1 stand for what?

Was the NLR determined in same fashion in all trials?

95%CI lack always the second value.

Figure 2 shows a non significant finding but the authors report it as significant in the text. The CI includes clearly the 1.

I would suggest to summarize the findings for Relapse and EDSS in a short paragraph as they come

Reviewer #2: In this study Cabanillas-Lazoi et al. describe in a meta analysis the role of neutrophil-to-lymphocyte ratio (NLR) in Neuromyelitis Optica Spectrum Disorder.

The analysis include six studies with 1036 patients in the analysis finally.

The main result is that NLR is increased in NMOSD.

In my opinion in the eligibility criteria the pre-treatment period in NMOSD patients should described in more detail as this period is important for the clinical impact.

Another limitation is that most studies were conducted in Chinese populations. This limits the general transferability. This should be clarified in the discussion.

Overall, the manuscript is well written and straight forward. The question is simple and precise answered. The topic is relevant for clinical neurology.

6. PLOS authors have the option to publish the peer review history of their article (what does this mean?). If published, this will include your full peer review and any attached files.

Reviewer #1: No

Reviewer #2: **Yes: **Jeremias Motte

---

## [Author Response · Author response to Decision Letter 0]

23 Dec 2022

1. Is the manuscript technically sound and do the data support the conclusions?

The manuscript was rewritten with the editors' corrections.

2. Has the statistical analysis been adequately and rigorously performed?

The statistical analysis was reevaluated in the results according to the editors' comments.

3. Have the authors provided all the data on which the conclusions of their manuscript are based?

We rewrote our results with the editors' comments.

4. Is the manuscript presented in an intelligible form and written in standard English?

We proceeded with a new and improved English translation of our manuscript.

5. Reviewer #1: 

5.1. Abstract :

The methods in the abstract were more specifically worded, the search terms, time and GRADE were added, and the GRADE had a more adequate description. In addition, the wording was corrected with respect to the comparison groups (Pages 2-3). 

5.2. Introduction:

The wording of the Kitley et al study and the paragraph regarding AQP-4 was corrected. Further expanded the relationship of NLR as a marker in other diseases.(Pages 3-4).

5.3. Methods: 

- Added search terms to the methods section.(Page 5).

- Detailed that there were no restrictions in the search, regarding time or language (Page 5).

- It was detailed that this work had a "peer review", referring to the fact that the analysis and selection of studies were made by 2 authors. (Page 6)

5.4. Results:

- It should be noted that our measure of effect is the difference in means, so the point of no effect is zero.

- The section on the extraction of the NLR values of the comparator was improved (Pages 5-7).

- We improved the wording to explain the definition of severe and mild NMOSD, as well as the use of EDSS criteria (Page 6).

- Improved wording was made on the follow-up of crude and adjusted association measures (Page 7).

- Improved wording on standard deviation extraction (Page 8).

- Improved statistical approach regarding random effects models and the use of the mean NLR difference as the outcome. (Page 8)

- Added NLR values in Table 1 for the NMOSD population as well as the controls. (Page 11-12)

- Detailed how the NLR was collected across all studies and the 95% CIs for each study. (Page 8)

6. Reviewer #2: 

6.1. Described in more detail the eligibility criteria as well as the pretreatment period (Page 5) 6.2.

6.2. The fact that most of the studies were conducted in Chinese populations, which limits the overall transferability, was redacted in limitations (Page 19).

---

## [Decision Letter · Decision Letter 1]

17 Jan 2023

Clinical and imagenologic significance of neutrophil-to-lymphocyte ratio in neuromyelitis optica spectrum disorder: a systematic review with meta-analysis

PONE-D-22-23904R1

Dear Dr. Alva Diaz,

We’re pleased to inform you that your manuscript has been judged scientifically suitable for publication and will be formally accepted for publication once it meets all outstanding technical requirements.

Kind regards,

Ralf A. Linker, MD

Academic Editor

PLOS ONE

Additional Editor Comments (optional):

Reviewers' comments:

Reviewer's Responses to Questions

**Comments to the Author**

1. If the authors have adequately addressed your comments raised in a previous round of review and you feel that this manuscript is now acceptable for publication, you may indicate that here to bypass the “Comments to the Author” section, enter your conflict of interest statement in the “Confidential to Editor” section, and submit your "Accept" recommendation.

Reviewer #1: All comments have been addressed

Reviewer #3: All comments have been addressed

2. Is the manuscript technically sound, and do the data support the conclusions?

Reviewer #1: Yes

Reviewer #3: Yes

3. Has the statistical analysis been performed appropriately and rigorously? 

Reviewer #1: Yes

Reviewer #3: Yes

4. Have the authors made all data underlying the findings in their manuscript fully available?

Reviewer #1: (No Response)

Reviewer #3: Yes

5. Is the manuscript presented in an intelligible fashion and written in standard English?

Reviewer #1: (No Response)

Reviewer #3: Yes

6. Review Comments to the Author

Reviewer #1: The authors have improved the paper importantly. Here are some minor points to address before publication.

Abstract :

Plese define more precisly the two values for the OR in the brackets. Is this a range or are these two values for example representing the OR of EDDS < or > 4 ? A standard presentaton would also include the 95% CI.

Methods

EDSS ranges from 0 to 10.

Results

Please review the brackets in the following sentence: Regarding subgroup analysis between studies with matched patients and controls, we observed that NMOSD patients had a higher NLR than healthy controls in matched studies (MD: 1.18; 95% CI: 0.94;1.43; I2 = 86%) (MD: 0.81; 95% CI: 0.56; 1.06; I2 = 0%)

Reviewer #3: In my opinion the study is interesting. The results are suitable for publication due to the simplicity of NLR application in clinical practice.

The meta-analysis presented here increased the evidence for this application in routine clinical practice.

7. PLOS authors have the option to publish the peer review history of their article (what does this mean?). If published, this will include your full peer review and any attached files.

Reviewer #1: No

Reviewer #3: No

---

## [Editor Report · Acceptance letter]

30 Jan 2023

PONE-D-22-23904R1 

Clinical and imagenologic significance of the neutrophil-to-lymphocyte ratio in neuromyelitis optica spectrum disorder: A systematic review with meta-analysis 

Dear Dr. Alva-Díaz:

I'm pleased to inform you that your manuscript has been deemed suitable for publication in PLOS ONE. Congratulations! Your manuscript is now with our production department. 

Kind regards, 

on behalf of

Dr. Ralf A. Linker 

Academic Editor

PLOS ONE